# Medication Adherence among Allogeneic Haematopoietic Stem Cell Transplant Recipients: A Systematic Review

**DOI:** 10.3390/cancers15092452

**Published:** 2023-04-25

**Authors:** Chiara Visintini, Irene Mansutti, Alvisa Palese

**Affiliations:** 1Department of Biomedicine and Prevention, University of Rome Tor Vergata, 00133 Rome, Italy; chiara.visintini@students.uniroma2.eu; 2School of Nursing, Department of Medical Sciences, University of Udine, 33100 Udine, Italy; irene.mansutti@uniud.it

**Keywords:** clinical pharmacist, graft-versus-host disease, haematopoietic stem cell transplantation, immunosuppressants, medication adherence, medication non-adherence, mortality, oral therapy, risk factors

## Abstract

**Simple Summary:**

Recipients of a haematopoietic stem cell transplantation must follow a complex treatment regimen that could reduce medication adherence (MA). Updated prevalence rates of MA, as well as factors promoting or hindering it and its outcomes, have not been summarised. Therefore, the primary aim of this review was to summarise the available oral MA prevalence data among adults who have received an allogeneic transplant and the tools used to measure it. The secondary aims were to find predictors and risk factors of medication non-adherence (MNA), the effectiveness of interventions, and the clinical outcomes of MNA. The MA is still an issue among these patients who report suboptimal prevalence rates. More than one measurement method should be considered when planning studies regarding MA. Additional research is needed to investigate other risk factors of MNA and to develop multidisciplinary interventions to improve MA, including the role of the caregivers’ and patients’ perceptions and MNA outcomes. This endeavour would produce more robust evidence to inform clinical practice.

**Abstract:**

Recipients of a haematopoietic stem cell transplantation (HSCT) may experience issues in medication adherence (MA) when discharged. The primary aim of this review was to describe the oral MA prevalence and the tools used to evaluate it among these patients; the secondary aims were to summarise factors affecting medication non-adherence (MNA), interventions promoting MA, and outcomes of MNA. A systematic review (PROSPERO no. CRD42022315298) was performed by searching the Cumulative Index of Nursing and Allied Health (CINAHL), Cochrane Library, Excerpta Medica dataBASE (EMBASE), PsycINFO, PubMed and Scopus databases, and grey literature up to May 2022 by including (a) adult recipients of allogeneic HSCT, taking oral medications up to 4 years after HSCT; (b) primary studies published in any year and written in any language; (c) with an experimental, quasi-experimental, observational, correlational, and cross-sectional design; and (d) with a low risk of bias. We provide a qualitative narrative synthesis of the extracted data. We included 14 studies with 1049 patients. The median prevalence of MA was 61.8% and it has not decreased over time (immunosuppressors 61.5% [range 31.3–88.8%] and non-immunosuppressors 65.2% [range 48–100%]). Subjective measures of MA have been used most frequently (78.6%) to date. Factors affecting MNA are younger age, higher psychosocial risk, distress, daily immunosuppressors, decreased concomitant therapies, and experiencing more side effects. Four studies reported findings about interventions, all led by pharmacists, with positive effects on MA. Two studies showed an association between MNA and chronic graft-versus-host disease. The variability in adherence rates suggests that the issues are relevant and should be carefully considered in daily practice. MNA has a multifactorial nature and thus requires multidisciplinary care models.

## 1. Introduction

Haematopoietic stem cell transplantation (HSCT) is a life-saving procedure for different acquired or inherited disorders, benign or neoplastic diseases of the haematopoietic system, and autoimmune or metabolic diseases [1,2,3]. In 2020, the first year of the COVID-19 pandemic, caused by severe acute respiratory syndrome coronavirus 2 (SARS-CoV-2), the number of HSCTs decreased compared with the previous year, with 45,364 HSCTs (a 6.5% decrease) across Europe in 41,016 patients (18,796 [41%] allogeneic, a 5.1% decrease, and 26,568 [59%] autologous, a 7.5% decrease), due also to the limited availability of staff and services [4]. These patients are routinely cared for in bone marrow transplant centres and, at their discharge, are required to adhere to a complex medication regimen based on prophylactic and immunosuppressive medications [5,6,7,8,9]. They must also follow social restrictions, daily health practices, and lifestyle changes [9,10,11,12], particularly during the first 100 days. Adhering to prescribed medications ensures the best outcomes and prevents complications, such as graft-versus-host disease (GvHD) [6,13,14,15]. However, the prevalence rates of adherence, factors promoting/hindering it, and its outcomes have not been summarised to date. Providing clinicians with a summary of the available evidence may increase their awareness of the issue, as well as its daily assessment and management, ultimately ensuring better patient outcomes.

The term adherence was first introduced as a Medical Subject Heading (MeSH) term in 1993. The World Health Organization (WHO) [16] defines it as ‘the extent to which a person’s behaviour—taking medication, following a diet, and/or executing lifestyle changes—corresponds with agreed recommendations from a health care provider’, presuming the patient agrees with the medical recommendations [16,17]. Medication adherence (MA), a MeSH term introduced in 2009, was conceptually developed in a framework composed of three elements: (a) the initiation, as the beginning of the prescribed medication intake; (b) the implementation, as the correspondence between the patient’s actual dosing and medical prescription, from initiation until the last dose is taken; and (c) the persistence, as the time between the first and the last intake of the prescribed medication [18]. The concept of medication non-adherence (MNA) was also established as ‘the late or non-initiation of the prescribed treatment, sub-optimal implementation of the dosing regimen, or early discontinuation of the treatment’ [18]. Different methods have been introduced to assess MNA, including subjective, objective, and biochemical measures [17], with self-report approaches often combined with objective methods [16,17].

Among HSCT recipients, only a systematic review [13] summarised the data regarding MA from five studies published up to 2014 on the paediatric and adult populations. The authors reported a range from 33% to 94.7% and a decline over time. A further prospective survey among 200 outpatient adult recipients of an allogeneic HSCT was conducted more recently in 2020 by Ice et al. [6]: 37.9% of patients reported MNA to oral immunosuppressors, while 51% showed MNA to non-immunosuppressors. Similarly, a cross-sectional multicentric French survey was performed in 2021 including 203 adult allogeneic transplanted patients by reporting a MNA rate of 75% [5]. A secondary data analysis published by Gresch et al. [14] among long-term allogeneic survivors reported 68.7% of MNA to immunosuppressors, of whom 62.2% did not have or had mild chronic GvHD and 80.2% had moderate or severe chronic GvHD.

The WHO has provided a five-factor classification of poor adherence in chronic illnesses [16], including socioeconomic, health system/healthcare team-, disease-, therapy-, and patient-related factors. While some factors have been associated with MNA in individuals with solid transplantations—such as youth, the male sex, anxiety and depression [19,20,21], and poor support [20]—the phenomenon among adult recipients of a HSCT remains scarcely investigated. Patients have been reported to experience anxiety and uncertainty in the transition from hospital to home [22], which could affect MA. Lower caregiver task efficacy and higher patient educational levels have also been documented to affect MNA 8 weeks after hospital discharge [9], as well as a younger compared with older age [6], with an increased risk for MNA in children than adults [13]. To promote MA, in their systematic review, Zanetti and colleagues [23] reported that a clinical pharmacist in the inpatient and outpatient settings may facilitate MA by managing pharmacotherapy-related problems, by discussing with the clinical team and by actively promoting medication reconciliation. This view has also been highlighted in more recent studies [7,24,25,26]. eHealth [27] has also been investigated in the Swiss SteM-cell-transplantatIon faciLitated by the eHealth (SMILe) Project [28,29,30], where an integrated model of care will be tested (eHealth and a nursing care coordinator) in a hybrid effectiveness-implementation trial, with a focus on MA [31].

MNA may result in different outcomes. There have been conflicting findings [13] regarding the incidence of GvHD, infections (viral or bacterial) hospital readmissions, and mortality [6,8,11,14,32]. Mishkin et al. [11] used the Stanford Integrated Psychosocial Assessment for Transplant (SIPAT) as a predictive tool of MNA in 85 recipients of an HSCT and reported no associations between SIPAT ratings and survival; however, patients with high-risk SIPAT scores had an increased risk of being admitted in the intensive care unit. 

However, despite the studies available, to our best knowledge, no comprehensive and up-to-date review has been published focusing on MA rates, predictors, and outcomes since 2017 [13]. Therefore, the primary aim of this systematic review was to summarise the oral MA prevalence data among adult recipients of an allogeneic HSCT and the tools used to measure it. Our secondary aims were to summarise the factors influencing MNA, the interventions investigated to date as capable of increasing MA, and the outcomes influenced by MNA.

## 2. Materials and Methods

We performed a systematic review registered at the International Prospective Register of Systematic Reviews (PROSPERO) (no. CRD42022315298). The review protocol has been developed and published [33]. Methods and findings have been reported according to the Preferred Reporting Items for Systematic Reviews and Meta-Analysis (PRISMA) [34] (see Appendix A).

### 2.1. Selection Criteria

We followed the Population, Intervention, Comparison and Outcomes and Time [PICO(T)] framework, as recommended by the Joanna Briggs Institute (JBI) methodology [35]. Eligible studies included adult patients who had undergone allogeneic HSCT, for any transplantation indications, and who had to take oral medications (both immunosuppressors and non-immunosuppressors, such as prophylactic medications) in the post-transplant phase up to 4 years after HSCT, according to the longest median follow-up among the available studies [33]. There were included primary studies on humans (a) published in any year; (b) written in any language; (c) with an experimental, quasi-experimental, observational, correlational, descriptive, cross-sectional, or longitudinal study design; and (d) reporting a low risk of bias in the quality assessment.

We excluded the following: secondary and qualitative studies, publications without original data (e.g., comments, letters to the editor and editorials), study protocols, including the paediatric population (<18 years old) but that did not stratify the results by paediatric and adult patients, or studies concerning autologous HSCT. Moreover, we excluded studies that assessed MA beyond the fourth year after HSCT or that considered adherence to some other factor (e.g., diet regimen or physical exercises).

### 2.2. Study Selection and Search Strategy

We investigated the PROSPERO database to check whether there were ongoing systematic reviews regarding this topic. Then, we searched the Cochrane Library, the Cumulative Index of Nursing and Allied Health (CINAHL), the Excerpta Medica dataBASE (EMBASE), MEDLINE via PubMed, and PsycINFO, and Scopus. Following the JBI methodology [35], the first author (CV) performed an initial search in the MEDLINE database (via PubMed) to identify keywords and MESH terms. The other authors (IM and AP) then refined and tested this search strategy. Finally, we applied the search string to the other databases.

We conducted the search in April 2022 and updated it on 10 May 2022, as reported in Appendix A. We also searched grey literature to check the availability of dissertations and reports or guidelines from international scientific societies. We searched for and carefully read conference posters and/or abstracts of presentations to determine if the original research articles had been published and to include them. We hand searched the reference lists of the included articles and of the excluded reviews to ensure comprehensiveness of the process. For articles with missing data, we contacted the corresponding authors to access the original data. For each contacted author, we sent one request via e-mail and, if we received no response, we sent a follow-up after 2 weeks.

We imported the titles and abstracts identified by the search strategy into EndNote® citation manager. CV and IM independently reviewed this information to determine if they were potentially eligible. Disagreements between the researchers were solved using consensus and consultation with AP. CV and IM then independently assessed the full texts to determine whether the articles could be included in the systematic review, with consultations with AP when necessary. The reasons for exclusion in this phase [36,37,38] are included in Appendix A. The full process is summarised in Figure 1.

### 2.3. Data Extraction

CV and IM independently developed a pilot grid to extract data for at least 10% of the included articles. Then, we established the final extraction grid, extracted the data, and double-checked the process. Extracted were the following data: (a) the main study characteristics (author(s), publication year, country, purpose(s), study design, setting, and duration); (b) the main characteristics of the population (the number, median age, gender distribution, race/ethnicity, haematological disease(s), type of allogeneic HSCT, time from HSCT); and (c) the frameworks used to identify MA and oral medication(s) (name of the immunosuppressors or non-immunosuppressors).

According to the aims of the systematic review, we distinguished the primary and secondary outcomes. The primary outcomes were the tool used to screen MA/MNA, the prevalence rate of oral MA/MNA, the provider(s) and the timing of MA measurement. The secondary outcomes were: (1) factor(s) affecting MA and MNA as predictor(s), risk factor(s), or facilitators/barrier(s)—in this context, also screening tool(s) assessing the risk of MNA; (2) intervention(s) to improve MA, the timing of provision and provider(s); and (3) outcome(s) of MNA as the incidence rate of infections (bacterial, viral or fungal), acute or chronic GvHD, hospital readmissions, disease relapse and/or mortality and time in days from HSCT.

For each study, we have summarised the main results narratively. CV and IM discussed discrepancies in data extraction until reaching a consensus, with consultations with AP if necessary.

### 2.4. Quality Assessment and Confidence in the Cumulative Evidence

CV performed a risk of bias assessment of all included studies and then IM cross-checked it; they solved discrepancies by consulting AP. The assessment was based on the JBI Appraisal Checklists [39,40,41], considering their capacity to guide the development of high-quality systematic reviews and address policy and practice interventions [35]. The JBI has a specific checklist for each study design (e.g., quasi-experimental, cohort, case-control, analytical cross-sectional and reporting prevalence data studies) [39,40,41]. For study designs not directly matching one of the categories defined by the JBI checklists, we first independently and then collectively matched the studies to the most suitable checklist among those proposed by the JBI (e.g., we considered a longitudinal study to be a single cohort study).

For every item included in the checklist, the possible responses were ‘yes’, ‘no’, ‘unclear’, and ‘not applicable’, with the overall appraisal resulting in ‘seek further info’, ‘include’ or ‘exclude’ [39,40,41]. Appendix A provides the results of the risk of bias assessment for each study; all the studies demonstrated sufficient quality, and thus we included them in the review. In analytical cross-sectional studies and cohort studies, failures in reporting mainly regarded the identification of confounding factors and the strategies implemented to deal with these confounding factors. Among the quasi-experimental studies and the studies reporting data prevalence, no data failures emerged.

Although in the study protocol [33] we stated that we would assess the quality of the evidence by using the Grading of Recommendations Assessment, Development and Evaluation framework [42], this was not possible due to the heterogeneity of the included studies with respect to the primary and secondary outcomes.

### 2.5. Data Analysis and Synthesis 

CV and IM independently summarised the findings. We have provided a qualitative narrative synthesis of the results due to the heterogeneity that emerged across studies. First, we have described the data collection tools and synthesised the prevalence rates of MA, divided by immunosuppressors and non-immunosuppressors, by providing the range (minimum to maximum values), mean, and median. Second, we have developed a description of the factors affecting MA and MNA according to the five-factor WHO framework [16] by summarising the predictors, risk factors, facilitators/barriers, and screening tools of MNA. Finally, we have described the interventions to improve MA and their providers, with trends of effectiveness (as outcomes improved, worsened, or were not statistically significant) and absolute and relative frequencies of the identified complications of MNA, reporting measures of associations where available.

## 3. Results

### 3.1. Main Study Characteristics

We included 14 studies (Figure 1). Their main characteristics are summarised in Appendix A. Among them, five (35.7%) are cross-sectional [5,6,11,12,14], four (28.6%) are quasi-experimental studies [7,24,25,26], two (14.3%) are longitudinal [9,15], two are observational studies [8,32] and one (7.1%) is correlational [10]. The studies had been conducted mainly in Europe (*n* = 6, 42.8%) [5,7,8,14,15,32] and in the US (*n* = 5, 35.7%) [6,9,10,11,12]. The oldest study dated back to 1993 [10], while eight studies (57.1%) had been published since 2020 [5,6,7,8,9,15,25,26]. The majority (*n* = 8, 57.1%) covered a study duration of ≤1 year [5,6,8,14,24,25,26,32], although two studies (14.3%) did not report this data [9,10].

In the 14 included studies, there were a total of 1161 individuals: 1049 adult HSCT recipients, from 23 [24] to 203 [5] transplanted patients per study, and 112 caregivers. The mean patient age ranged from 38.1 years [10] to 56.6 years [6], except for two quasi-experimental studies: in the first [25], most patients in both the experimental and control group were <65 years old (76% and 69%, respectively); in the second study [26] authors reported the mean age of both adult and paediatric patients. Overall, the male sex was predominant (*n* = 639, 60.9%). Ethnicity was not reported in 10 studies [5,7,8,11,14,15,24,25,26,32]; among the remaining studies, Caucasians were the most represented (*n* = 242, 65.6%).

Acute leukaemia (*n* = 437, 52.9%) was the haematological disease investigated most often; two studies (14.3%) did not provide this information [6,24]. Two studies (14.3%) [10,11] combined allogeneic and autologous HSCTs; however, among allogeneic HSCTs, most were matched unrelated donor (MUD) (*n* = 279, 47.4%), followed by 246 (41.8%) HLA-identical and 41 (7.0%) haploidentical HSCTs (data missing from five studies [5,15,24,25,32]). The mean time between HSCT and the study outcome assessment ranged from 72 days [32] to 3.9 years [14]; three studies did not report this information [9,24,25].

### 3.2. Oral MA: Assessment Tools and Rates

Only four studies (28.6%) reported the theoretical framework used to assess MA [5,8,11,14]. MA regarding oral immunosuppression therapy had been examined in three studies [7,14,15], whereas seven studies [5,6,8,9,12,24,32] had assessed both immunosuppressor and non-immunosuppressor adherence to anti-infective prophylaxis. Four studies assessed MA; however, they did not provide the name of the prescribed oral medications [10,11,25,26].

The studies used several different tools used to measure MA (Table 1) with the predominance of subjective measures (78.6%) such as self-report questionnaires administered by professionals. They were used mainly in their original version, namely the Morisky Medication Adherence Scale (MMAS) with eight items [6,32], the Compliance Evaluation Test (CET) with six items [5], the Basel Assessment of Adherence to Immunosuppressive Medication Scale (BAASIS) with six items [14], the Immunosuppressant Therapy Adherence Scale (ITAS) with five items [6], the Medication Experience Scale for Immunosuppressants (MESI) with seven items [15], and the Brief Medication Questionnaire (BMQ) with 11 items [26]. Some studies also adapted tools from their original versions, such as the Health Habits Assessment (HHA) with two questions related to MA [9,12], the four-item MMAS [24], and the five-item CET [5]. Subjective measures were used also to assess MA before HSCT, using the Self-Rating of Pre-HSCT Adherence (SRPA) [10].

Among the objective measures (35.7%), the number of dispensation and refill records was used the most [6,8,24]; biochemical measures (21.4%), such as measurement of drug serum levels of immunosuppressors [6,7,8] and the number of serum assays [7], were used the least.

The tools were used alone (e.g., the MMAS [32]) or combined: Ice et al. [6] used the MMAS for non-immunosuppressors and the ITAS for immunosuppressors, in addition to the number of prescription refill records and serum drug levels. On the other hand, Gresch et al. [14] used more subjective methods: the BAASIS, a physician dichotomous evaluation based on the drug serum level and a third and final evaluation combining the BAASIS score with the physician’s evaluation.

Although three studies [8,14,32] did not report the time, MA was assessed during outpatient visits after hospital discharge, weekly from the second week and every 7–10 days up to six visits [24], from 3–5 weeks [25], at weeks 4 and 8 [9], and three times until 100 days [10] or 36 months [15] post-HSCT.

According to the measures adopted, the MA rate varied from 18.8% using the CET [5] to a mean decrease of 1.53 points based on the adapted MMAS after 6 weeks, reaching the score of 0 (corresponding to 100% of adherence) [24]. Polito et al. [25] using a visual analogue scale (VAS) from 0% to 100% reported a median MA rate of 100% at the follow-up 3–5 weeks after discharge. The median MA rate, considering all the rates (*n* = 26) collected with the different tools, was 61.8%. The mean rate was 55.9%, calculated as a weighted mean (the number of patients for each reported rate divided by the total of patients). In particular, the median and mean rates regarding HSCT patients taking immunosuppressors were 61.5% and 61.9%, respectively, (range 31.3–88.8%) [6,7,8,9,12,14,15,24] and, among those taking non-immunosuppressor therapy, they were 65.2% and 58.0%, respectively (range 48–100%) [6,8,9,12].

### 3.3. Factors Affecting MA/MNA

Based on the five-factor WHO classification of poor adherence [16], Figure 2 summarises the factors that emerged in 10 studies (71.4%) investigating MNA [5,6,9,10,11,12,14,15,26,32] and facilitators of MA [12,32]. Most are associated with the patient-related domain (age, race, distress, education level, psychosocial risk, and caregiver task efficacy) and the therapy-related domain (cyclosporine A [CsA] and valacyclovir/acyclovir treatment and the number of daily medications and/or intakes). Experiencing more side effects was also a risk factor for MNA. No factors emerged from the socioeconomic or healthcare team and system-in-place-related domains.

As summarised in Table 2, age was the most investigated predictive variable of MNA. In the cross-sectional study by Belaiche et al. [5], the non-adherent group had an average age of 50.4 years, significantly younger than the adherent group (55.1 years, *p* = 0.04), as also confirmed in the multivariate analysis. A cross-sectional study [6] also pointed out the role of younger age as a risk factor for MNA, where older age was associated with less MNA based on the MMAS (55.5 years in non-adherent patients vs 57 years in adherent patients, *p* = 0.009; odds ratio [OR] 0.97, *p* = 0.016). However, age was not considered a risk factor for adherence to immunosuppressors (data are not shown, based on the ITAS). The authors concluded by hypothesising a positive association between non-Caucasian race and MNA [6]. Moreover, Lehrer et al. [32] reported a positive correlation between age and adherence (Spearman’s correlation coefficient [*ρ*] = 0.47, *p* = 0.03).

Among the other patient-related factors, a higher education level of the patients (at or above a college degree) was associated with higher MNA at 8 weeks post-HSCT [9]. Moreover, severe distress was associated with MNA, both towards immunosuppressors (OR 1.17, *p* = 0.026, based on the ITAS) and non-immunosuppressors (OR 1.15, *p* = 0.035, based on the MMAS) [6]. The role of caregivers was also examined [9], indicating that their task efficacy at 4 weeks post-discharge was associated with MNA: in particular, MNA increased as the caregiver task efficacy decreased. A better-quality relationship between patients and their caregivers (Pearson’s correlation coefficient [r] = 0.52, *p* = 0.015) was correlated with higher immunosuppressor MA [12]. Even less stress in completing the medical regimen tasks, perceived by both patients (*r* = −0.45, *p* = 0.042) and caregivers (r = −0.57, *p* = 0.011), was a facilitator of adherence to immunosuppressors [12]. Nevertheless, adherence to other medications was correlated with less perceived stress of the medication regimen only by the caregivers (r = −0.46, *p* = 0.048) [12].

Regarding the therapy-related factors, CsA (57.8% vs 69.0%) and valacyclovir/acyclovir (89.6% vs 73.6%) treatment, in addition to experiencing more side effects (46.3% vs 22.3%) (*p* = 0.023, *p* = 0.023, and *p* = 0.009, respectively) affected MNA [5]. An increase in the number of daily oral immunosuppressors was also associated with MNA [14]. Similarly, a higher number of as-needed medications increased the odds of MNA [6]. Otherwise, an increase in the number of daily intakes based on the five-item CET was associated with MNA, with results not confirmed in the multivariate analysis (OR 1.138, *p* = 0.32) [5]. A lower number of concomitant medications was also associated with MNA (OR 0.85, 95% confidence interval [CI] 0.74–0.98, *p* = 0.024) [14]. However, Ice et al. [6] reported a decrease in MNA based on the MMAS per an increase in one scheduled medication daily (OR 0.92, 95% CI 0.86–0.998, *p* = 0.043). On the other hand, Belaiche et al. [5] found no differences with respect to the number of taken medications.

#### MNA Screening Tools

Three studies documented findings in this field [10,11,15]. First, as a pre-HSCT screening tool of MNA, Mishkin et al. [11] assessed the SIPAT—already validated among the solid organ transplanted population—to predict MNA retrospectively in 50 allogeneic and 90 autologous HSCT patients. Adjusting for allogeneic vs autologous, MNA after HSCT showed an increase per one point of the SIPAT score (higher psychosocial risk) (OR 1.162, *p* < 0.001) (Table 2). Specifically, a SIPAT score of ≥18 was the cut-off value to define ‘high risk’ of MNA (specificity of 89.6% and sensitivity of 55.6%). Differently, Scherer et al. [15] used the Transplant Evaluation Rating Scale (TERS) as a pretransplant psychosocial screening instrument to predict mortality, and secondarily, MA. A total of 61 allogeneic HSCT patients were prospectively included in a 3-year follow-up; higher TERS scores (higher psychosocial impairment) were associated with lower survival at 90 days post-HSCT (hazard ratio [HR] 2.399). However, TERS scores were not correlated with MA, measured with the MESI, both in the inpatient (at 3 months post-HSCT: *r* = −0.026, *p* = 0.85; at 1-year post-HSCT: *r* = −0.206, *p* = 0.12) and in the outpatient setting (at 3 months post-HSCT: *r* = −0.009, *p* = 0.96; at 1-year post-HSCT: *r* = 0.03, *p* = 0.9), respectively. Finally, in a prospective observational study, Hoodin [10] tested the association between self-reported pre-HSCT adherence, using the SRPA, and MA post-HSCT based on the number of medication infractions; no relationship emerged (*ß* = −0.083, *t* = 0.586, *p* = 0.561).

### 3.4. Interventions to Improve MA

As reported in Table 3, four studies investigated interventions to improve MA, all focusing on clinical pharmacist–led interventions. Two were single-arm trials with a prospective cohort [24,26], one was a pre- and post-test study with an historical cohort [7], and one was a prospective pre- and post-cohort comparison study [25].

Specifically, Charra et al. [7] and Zanetti et al. [26] delivered pharmaceutical consultations to recipients of an allogeneic HSCT before discharge and then continued in the follow-up period, up to 100 days. Charra et al. [7] provided consultations to 61 recipients of an allogeneic HSCT starting the day before the discharge, while Zanetti et al. [26] provided it to 18 adults and 9 children from admission to the day before discharge. On the other hand, Chieng et al. [24] provided consultations to 23 patients only in the post-discharge phase, until the first month after HSCT. Polito et al. [25] were the only ones who delivered consultations solely before discharge, comparing a patient self-administration programme (*n* = 25) to a one-on-one counselling session (*n* = 26), performed by a pharmacist. No interventions were scheduled during the follow-up period.

Although Zanetti et al. [26] did not show the compliance prevalence data for the adults after the last pharmaceutical consultation, an improvement among the adult population can be hypothesised based on the general data presented (compliance score in adult and paediatric patients: 40.74% at baseline vs 70.37% at the last consultation; *p* = 0.0115). Similarly, Chieng et al. [24] reported that the weekly pharmaceutical consultations were associated with an improvement in MA with a mean decrease of 1.53 points on the MMAS (95% CI 1.12–1.94, *p* < 0.0001), reaching the maximum level of adherence (score 0) at the last consultation post-discharge.

Charra et al. [7] found a positive trend in favour of the prospective cohort that underwent pharmaceutical consultations compared with the retrospective cohort that did not receive any consultations; however, the finding was not significant (61.5% vs 53.0%, *p* = 0.07). Polito et al. [25] provided the intervention group with pharmaceutical consultations as well as nursing supervision in the self-administration of the medications. The median self-reported MA rates based on the VAS were similar between the groups (100%, *p* = 0.12).

### 3.5. Clinical Outcomes of MNA

Eight studies reported clinical outcomes [6,7,8,11,14,15,25,32] (we did not consider Zanetti et al. [26] because the authors did not show the outcome data for adults); however, only five of them [6,8,11,14,32] investigated the relationship with MNA (Table 2). The most investigated outcome was GvHD [6,8,14,32], followed by mortality [6,11], hospital readmissions [8], and infections [8]. None of the studies evaluated associations between MNA and disease relapse.

GvHD has been investigated in terms of acute (aGvHD) and chronic (cGvHD) manifestations. García-Basas et al. [8] reported different aGvHD rates, with a prevalence of grade 1 (40.9%) among patients who were adherent (*n* = 17, 45.9%) and non-adherent (*n* = 5, 55.6%) to immunosuppressors. MA emerged as a protective factor (OR 0.68, 95% CI 0.157–2.943), but the findings were not significant (*p* = 0.718). Likewise, in their pilot study, Lehrer et al. [32] showed a lower incidence of aGvHD among patients who were more adherent to medications (26.7% vs 38.9%), although this difference was not significant (*p* = 0.71). Differently from aGvHD, cGvHD has been documented as associated with MNA. Gresch et al. [14] reported incidence rates of cGvHD of 63.6% and 84.4% for adherent and non-adherent patients, respectively. Moreover, they found a positive association between MNA and higher grades of cGvHD (OR 3.01, 95% CI 1.27–7.14, *p* = 0.012). Ice et al. [6] reported similar results: overall, 161 patients (80.5%) developed cGvHD; mild GvHD was associated with MNA to immunosuppressors (OR 2.63, 95% CI 1.04–6.66, *p* = 0.042).

The relationship between MNA and mortality has been investigated in two studies [6,11]. Mishkin et al. [11] found no relationship between adherence and mortality. Moreover, in their prospective survey, Ice et al. [6] assessed the mortality among adherent and non-adherent patients to immunosuppressors and other medications. Although they identified trends towards lower survival in less adherent patients, the differences were not significant for immunosuppressors (HR 1.43, 95% CI 0.83–2.44, *p* = 0.19) and non-immunosuppressors (HR 1.33, 95% CI 0.72–2.44, *p* = 0.36).

García-Basas et al. [8] reported that 18.8% of hospital readmissions were associated with GvHD and concerned only adherent patients. Similarly, infections or fevers that required hospitalisation represented three-quarters of the overall readmissions. Overall, 67.4% of the transplanted patients presented with fever or infection; however, it was not possible to compare the incidence of these events among adherent vs non-adherent patients, as all patients were classified as adherent to anti-infection prophylaxis [8].

## 4. Discussion

To the best of our knowledge, this is the most recent systematic review aimed at summarising the evidence regarding oral MA among adult HSCT recipients. Many steps have been taken in the 20 years since the mini-review by Bishop et al. [44] in 2002 and the last review by Morrison et al. [13] in 2017, in which the authors called for research action. In the last 20 years, more than two studies have been published per year, with mainly cross-sectional and quasi-experimental designs, suggesting the need to again call for action in this field with limited evidence, by designing more robust studies capable of informing clinical practice.

The USA and European countries are the leaders in this research field; extending the research worldwide, considering the relevance of culture on MA, is strongly suggested. On average, a higher number of patients has been included in each study (*n* = 1049) as compared to that reported by Morrison et al. [13] (*n* = 277). Their mean age is in line with that documented in this transplanted population [45]. Acute leukaemia and allogeneic MUD were the more common indications for and performed types of HSCT, all in line with the indications for and types of transplants mostly performed in Europe [4,46].

### 4.1. Oral MA: Assessment Tools and Rates

The theoretical framework of MA was not reported in most studies, suggesting the need to promote conceptual clarity that may also increase the homogeneity across studies and the comparability of the findings. Moreover, identifying the specific points where the patients report MA difficulties, using an established MA framework [18], allows targeted actions to support adherence over time.

The variety of assessment methods, mostly validated self-report questionnaires, prevented us from comparing MA rates across studies. The WHO [16] recommends using self-report approaches in combination with objective methods, given that self-report methods alone may result in overestimations [13] and do not have the capacity to distinguish intentional vs non-intentional MNA. On the other hand, objective measures might also have some limitations [13,16]. The use of electronic monitoring systems requires a special container for the drug with an integrated chip on the lid that records each opening of the container, but this count fails to inform whether the pills are actually taken [13,16]. Moreover, monitoring the drug serum assays should consider the patients’ metabolisms and the potential medication interactions, as well as the related costs [16]. Thus, more research is needed to establish the most reliable systems to measure MA, also regarding self-report measures, given that the best tool in terms of validity and reliability [46] has not yet been established.

The documented MA prevalence rates (median 61.8%, range 18.8–100.0%) are higher compared with that documented by Morrison et al. [13], where the overall adherence to the daily medication regimen was 33–94.7%. However, the participants involved in studies tend to be more adherent than in the real world [5], suggesting that these findings could be overestimations. With respect to immunosuppressors, the data (31.3–88.8%) are partially in line with MA rates for immunosuppressors in patients who have received kidney (45–64%) [21] and heart (25–80%) transplants [20] and the previous solid organ transplantation literature (64–97%) [47]. The wide range of adherence rates could be explained by the differences in the average time between HSCT and the study outcome assessment (72 days [32] to 3.9 years [14]), in addition to the different assessment methods used.

According to the findings, three main trends have emerged in MA over time: (1) there has been a slight decrease in MA from 88.8% to 84.3% based on the ITAS (for immunosuppressors) and from 65.2% to 61.4% based on the MMAS (for non-immunosuppressors) at weeks 4–8 post-discharge [9]. (2) Differently, MA has been reported to be stable across three repeated measures using the number of self-reported medication infractions as an adherence measure until 100 days after HSCT [10]. (3) MA increased from 40.5% to 50.0% based on the MESI from 3 to 36 months after HSCT [15], and decreased by 1.53 points (reaching the score of 0, corresponding to 100% adherence) based on the adapted MMAS at 6 weeks [24]. However, Chieng et al. [24] limited the outcome assessment to a few weeks; thus, the white-coat adherence (improvement in adherence just before a medical appointment [48]) may have affected the findings. This potentiality suggests that MA could be a time-dependent variable and its assessment should be investigated relative to time. Therefore, prospective observational and longitudinal studies are needed, with careful and accurate reporting of the time of MA measurements.

### 4.2. Factors Affecting MA/MNA and Screening Tools

Ten studies investigated the risk factors of MNA or the facilitators of MA documenting mainly patient-, therapy-, and disease-related factors [16]. Younger age was associated with an increase in MNA [5,6,32], as also reported among heart and kidney-transplanted patients [20,21]. No studies found an association between sex and MNA, while only one [9] investigated the role of education, suggesting the need to investigate more, considering that the role of education on MA has reported conflicting findings [49]. The fact that MNA was associated with a higher patient education level may be explained from different angles, as the higher distress or psychosocial risk perceived due to the full understanding of the health situation; however, these hypotheses did not emerge from the study by Posluszny et al. [9].

The role of the dyad ‘patient and caregiver’ has also been investigated in two studies [9,12], suggesting an important influence of the caregiver’s support perceived by the patient, as also previously documented [50], in which patients with a dedicated caregiver had a higher probability of surviving than patients without a caregiver. Therefore, strategies to assess the quality of the support perceived by the patient and his/her caregiver are required. Moreover, psychological aspects are important, as previously documented [43,44] also in other fields, where perceived barriers such as depression, anxiety, negative personality, substance abuse, and worse psychosocial functioning [20,21] resulted in MNA among patients who have received a heart or kidney transplant. However, in our review, only severe distress emerged as associated with MNA to immunosuppressor and non-immunosuppressor therapy [6].

Conflicting findings have emerged regarding the therapy-related factors. Gresch et al. [14] found an association between MNA and a higher number of daily immunosuppressive pills, where the unpleasant smell and taste of pills [5] or the onset of cGvHD of the mouth mucosa may have led to a more difficult taking of the medications prescribed; Ice et al. [6] discovered an increase in the number of scheduled medications as associated with decreased odds of MNA. Patients taking more scheduled medications had more severe cGvHD and were thus more likely to adhere to their recommended medication regimen. Alternatively, an explanation could be that patients with fewer medications pay less attention to their medication management [14]. None of the studies investigated factors belonging to the socioeconomic and healthcare team dimensions, suggesting two additional areas of investigation.

Continuing to investigate factors related to MNA might also contribute to the identification and validation of screening tools to detect patients at risk early. In this field, limited evidence is available to date [10,11,15].

### 4.3. Interventions to Improve MA

All reported interventions that aimed to promote MA were led by pharmacists, except for the one proposed by Polito et al. [25], who introduced nursing supervision of the patients’ medication self-administrations. Although only one study [24] yielded significant results for adults, the delivery of pharmacist-led interventions showed positive effects on MA [7,25,26]. The interventions have been offered both during hospitalisation and after discharge. In the kidney transplant setting, several different interventions have been documented in addition to those led by pharmacists (e.g., educational–behavioural interventions, telephone calls, or online websites [21]). Thus, multicomponent, and multidisciplinary interventions should be investigated by also considering healthcare professionals’ and patients’ experiences. For example, a survey [51] among 143 European nurses reported that directly questioning patients about adherence, offering materials to read, and training them during their inpatient stay were the most frequently perceived effective interventions to promote MA. It would be interesting to implement nursing management models, such as those based on the clinical nurse specialist—with skills regarding MA—given the positive impact registered in some onco-haematological contexts [52]. On the other hand, as part of the SMILe Project [53], patients reported adopting strategies to manage the medication intake at home, such as the use of dispensers and electronic alarms, or linking them to their own routine (e.g., during meals). In one study, the medication event monitoring system (MEMS) button, a small device that must be pressed at each medication intake, was the most useful electronic method to monitor MA [53]. In addition, taking personal responsibility and adopting positive beliefs have been reported to be important for patients [54]. Therefore, multidisciplinary interventions capable of including all these aspects are required to increase evidence in the field of MA.

### 4.4. Clinical Outcomes of MNA

MNA to immunosuppressors showed interesting findings regarding GvHD [6,8,14,32]. Although the association between MNA and aGvHD was not statistically significant [8,32], non-adherent patients showed a high occurrence and higher grades of cGvHD [6,14]. However, it is challenging to establish the association between MNA and cGvHD because this condition could be influenced by genetic and pathophysiological components [6,14,55] that are difficult to control. Indeed, even the association between MNA and mortality has been hard to investigate; only one study [6] suggested lower survival among patients with immunosuppressor and non-immunosuppressor non-adherence. Mortality actually occurred mainly due to disease relapse, despite improvements in survival [56]. However, longitudinal studies searching for clinical outcomes of MNA, including mortality and disease relapse, are recommended.

García-Basas et al. [8] concluded that GvHD, infections, or fever were the major causes of hospital readmissions; however, all the readmitted patients were adherent to their medical regimens. Thus, an association between MNA and hospital readmissions could not be established, suggesting that this relationship must also be further investigated.

Finally, to define the threshold between MA and MNA, the studies are based on the indications of the adopted tools; usually, 80% of taken medication is the arbitrary threshold to consider a patient adherent or non-adherent [13,16]. However, this must be done with respect to clinical outcomes, as reiterated by Morrison et al. [13]. Therefore, researchers are invited to explore the possible relationships between scheduled deviations in adherence and poor clinical outcomes; they should determine how much of the variability in the outcomes is due to issues concerning immunosuppressive drug-taking behaviours. We also request that researchers make efforts to identify positive outcomes of MA, such as a possible impact on quality of life.

### 4.5. Limitations

This systematic review has several limitations. First, considering the multiple aims to develop a comprehensive guide regarding MA and MNA in this field, we may have missed some studies (e.g., those regarding specific factors affecting adherence as perceived by healthcare professionals). In addition, we used different search strings, in line with the aims of the review; however, the search terms we identified are general by design to ensure a comprehensive overview of all studies. In addition, we excluded some studies conducted in the paediatric setting, considering the differences with the adult population [13,57]. The inclusion of this population in future reviews might contribute to broadening the summary of the evidence produced in the field. Moreover, according to the study protocol, we did not search for an association between comorbidities and MNA. However, considering that comorbidities have been reported to influence MNA in other settings [58], this variable could be further investigated among the HSCT recipients.

Second, in the data analysis, we used both a deductive (e.g., the five-factor WHO framework [16]) and an inductive approach. When summarising the findings, the adoption of multiple approaches might have forced some interpretations (as in the case of the deductive approach); whereas, other interpretations might have better reflected the state of the research in the field (when we used an inductive approach) [59].

Third, although each author of this review has been engaged in this field for a long time (>1 year), our experiences and backgrounds might have influenced the interpretation process. 

Fourth, according to the heterogeneity in the included studies, we could not perform a quantitative synthesis of the results in the form of a meta-analysis. Additionally, the relationship between the effectiveness of interventions and trends has not been weighted according to study design, size, and effect dimensions. This is an area for potential improvement in future reviews.

## 5. Conclusions

The HSCT medication regimen is complex, involving multiple medications, based upon immunosuppressors and prophylactic therapy. The variability in adherence rates (18.8–100.0%), especially for immunosuppressors (31.3–88.8%), suggests that these issues are relevant and should be carefully considered in daily practice. Several instruments have been used to date, predominantly self-report questionnaires that involve the patients. Factors affecting MNA are mainly related to patients (e.g., younger age, higher psychosocial risk and distress, and lower caregiver task efficacy) and therapy (increase in daily immunosuppressive pills/intakes or decrease in concomitant medications). Despite its relevance and the presence of several modifiable factors, interventions improving MA have been focused on those led by pharmacists, with different trends of effectiveness. Therefore, there is the need to promote intervention studies by considering the factors documented and the complexity of MA. According to the findings, MNA is associated with cGvHD, while evidence regarding its association with infections, hospital readmissions, and mortality is limited.

According to the findings, we suggest the following practical implications at four different levels.

(1)Clinical level: we encourage the use of validated tools to assess MA longitudinally both for immunosuppressors and non-immunosuppressors by reporting findings in the clinical records, especially during the follow-up, and by involving caregivers as an integral part of the care process. Increasing awareness of these issues is important to address educational interventions, especially towards high-risk patients or those with MNA factors, e.g., young people.(2)Research level: it is necessary to use an MA framework and a multidimensional adherence measurement (subjective, objective, and biochemical), corroborating each measure with the others, in studies on MA. More emphasis should be given to validating MNA screening tools and on the multifactorial nature of MNA, considering also the socioeconomic-related and healthcare team and system-in-place-related factors. The role of caregivers’ and patients’ perceptions should also be considered. More attention should be devoted to MNA outcomes through prospective studies. In addition, when developing future MA tools, the threshold to define an adherent or a non-adherent patient should be based on outcomes.(3)Educational level: the MA topic should be addressed in undergraduate, postgraduate, and continuing education curricula, allowing an understanding of the differences in MA among HSCT, solid organ transplantation, and other chronic illnesses, by promoting multidisciplinary patient-centred care. Healthcare students at all levels should be aware of the role that they can play in the HSCT team. Adherence-related skills should be promoted and evaluated by also using peer education or simulation.(4)Management level: benchmarking adherence data across bone marrow centres must be encouraged, also to inform leaders in carrying out different models of care delivery, especially transitional care. Costs for adherence-enhancing interventions, including the effects of MNA such as GvHD treatment and readmissions, should be considered at the managerial level.

## Figures and Tables

**Figure 1 cancers-15-02452-f001:**
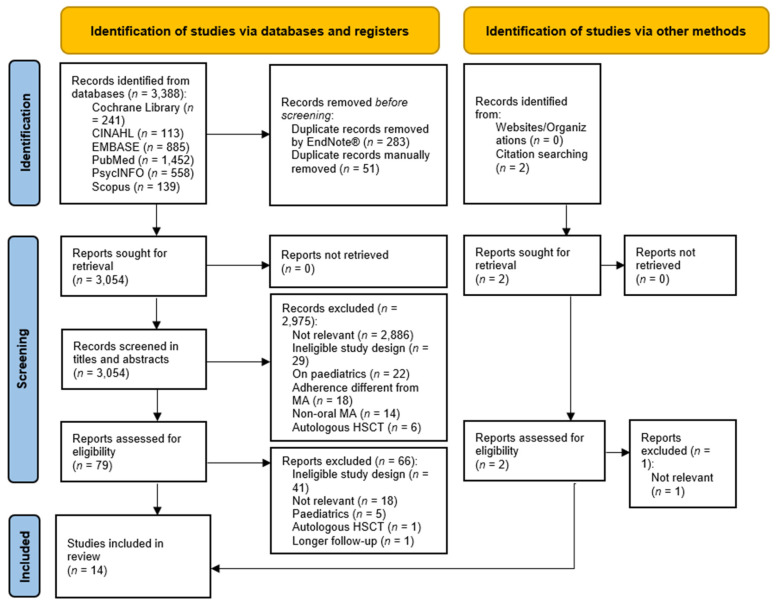
PRISMA flow diagram describing the process of the study included in the systematic review [34]. CINAHL: Cumulative Index to Nursing and Allied Health Literature; HSCT: haematopoietic stem cell transplantation; MA: medication adherence; *n*: number.

**Figure 2 cancers-15-02452-f002:**
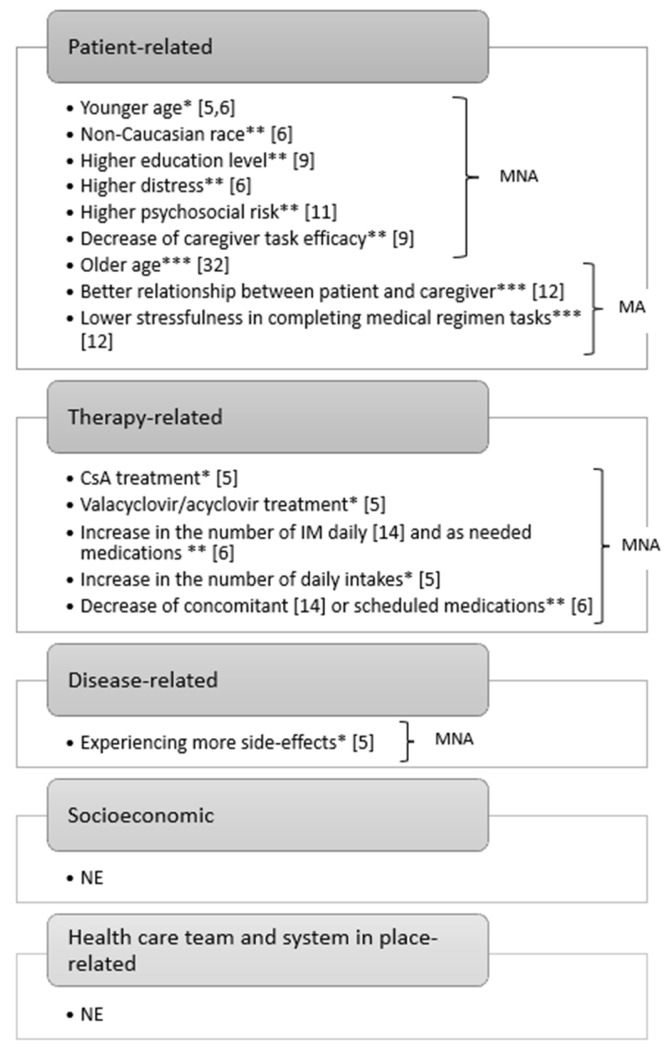
Summary of factors affecting MNA and MA among the adult allogeneic HSCT population, according to the five-factor WHO classification [16]. CsA: cyclosporine A; HSCT: haematopoietic stem cell transplantation; IM: immunosuppressors, MA: medication adherence; MNA: medication non-adherence; NE: not emerged; WHO: World Health Organization. * from univariate analysis. ** from multivariate analysis. *** using Pearson’s correlation coefficient [5,6,9,11,12,14,32].

**Table 1 cancers-15-02452-t001:** Measurement methods [16] to assess MA after HSCT from the 14 included studies.

Subjective (*n* = 13, 78.6% *)	Objective (*n* = 4, 35.7% *)	Biochemical (*n* = 2, 21.4% *)
BAASIS, validated [14]	Number of dispensation/refill records [6,8,24]	Drug serum level [6,7,8]
BAASIS in addition to physician evaluation [14]	Number of dose administration aids [24]	Number of serum assays [7]
BMQ, validated [26]	Pill counting [10]	
CET, validated and adapted [5]	Criteria based on an interdisciplinary team consensus [11]	
HHA, adapted [9,12]		
ITAS, validated [6]		
MMAS, validated [6,32] and modified [24]		
MESI, validated [15]		
Physician dichotomous evaluation based on drug serum levels [14]		
VAS, validated [25]		
24-h recall [10]		
Likert scale ** [15]		
SRPA **, validated [10]		

BAASIS: Basel Assessment of Adherence to Immunosuppressive Medication Scale; BMQ: Brief Medication Questionnaire; CET: Compliance Evaluation Test; HHA: Health Habits Assessment; ITAS: Immunosuppressant Therapy Adherence Scale; MA: medication adherence; MESI: Medication Experience Scale for Immunosuppressants; MMAS: Morisky Medication Adherence Scale; *n*: number; SRPA: Self-Rating of Pre-BMT Adherence; VAS: visual analogue scale. * % with respect to the 14 included studies. ** in the pre-HSCT assessment.

**Table 2 cancers-15-02452-t002:** Outcomes investigated from the included studies, organised according to our primary and secondary aims.

First Author, Year, Country	Prescribed Medication(s) Name, Frequency of Prescription among Population (*n*, %)	Outcomes Investigated				Main Results of the Study
		Primary Outcomes Metric of the Tool Used to Screen MA/MNA, Prevalence Rate of Oral MA/MNA, Provider(s) and Timing of MA Measurement	Secondary Outcome 1 Factors Affecting MA and MNA as Predictor(s), Risk Factor(s), Facilitators/Barrier(s) and/or Screening Tool(s) of the Risk of MNA	Secondary Outcome 2 Intervention(s) to Improve MA, Timing of Provision and Provider(s)	Secondary Outcome 3 Incidence Rate of Infections, GvHD, Hospital Readmissions, Disease Relapse and/or Mortality and Time in Days from HSCT due to MNA	
**Cross-sectional studies**						
Belaiche et al., 2020 [5], France	Immunosuppressives: -CsA = 157 (84.4%) -Corticosteroids = 66 (35.5%) -FK = 9 (4.8%) -mTOR inhibitors = 9 (4.8%) Anti-infective prophylaxis: -Valacyclovir/acyclovir = 170 (86.7%) -Trimethoprim/atovaquone = 154 (78.6%) -Amoxicillin/oracillin = 153 (78.1%) Antiemetics/antinauseants = 39 (20.6%)	CET, a validated six-item self-report questionnaire (*n* = 192) CET, adapted self-report five-item questionnaire (*n* = 192) Good MA = 36 (18.8%), moderate MA = 139 (72.4%), poor MA = 17 (8.9%) (CET six item) 56% take medications later than the usual 67% think they have to take too many tablets and 9% admit forgetting to take medication some days Good MA = 82 (41.5%), moderate MA = 115 (57.5%), poor MA = 3 (1.5%) (CET five item) Patients and collected by the medical or paramedical team Once, during hospitalisation or medical visit	Univariate analysis: (CET six-item) -Age < 50 years (50.4 vs 55.1, *p* = 0.041) -Treatment with CsA (87.3% vs 69.0%, *p* = 0.023) and valacyclovir/acyclovir (89.6% vs 73.6%, *p* = 0.023) -Experiencing more side effects (46.3% vs 22.2%, *p* = 0.009) (CET five item) -Age < 50 years (49.8 vs 53.5, *p* = 0.049) -More intakes per day (3.7 vs 3.3, *p* = 0.049) Multivariate analysis: (CET six item) -Age (*ß* = −0.0095, *p* = 0.053) (CET five item) -Age (OR = 0.97, *p* = 0.041)	-	-	According to the six-item CET, 81.3% were non-adherent and 18.8% were adherent, while considering the five-item CET 5, 58.5% were non-adherent and 41.5% were adherent Age < 55 years was the only factor associated with NA that emerged from the multivariate analysis
Gresch et al., 2017 [14], Switzerland	Immunosuppressors: -Steroids = 11 (11.6%) -CsA or FK = 48 (50.5%) -mTOR inhibitor or mycophenolate) = 5 (5.3%) combination + steroids = 31 (32.6%) -Not documented = 4 (4.0%) Mean number of immunosuppressive pills = 2.5 (1–12) Mean number of concomitant medications = 8.0 (1–22)	BAASIS, a validated six-item self-report questionnaire Physician dichotomous evaluation based on drug serum levels BAASIS scores combined with physicians’ evaluation MNA = 64 (64.6%): 33 (33.3%) had missed at least one dose 3 (3.2%) had missed at least two consecutive doses (‘drug holidays’) 61 (61.2%) had timing NMA (2 h too early or too late) 4 (4.1%) had themselves reduced the dosages 3 (3.1%) stopped the treatment early (non-persistent with therapy) MNA = 18 (18.9%) MNA = 68 (68.7%) Patients for self-report Laboratory A senior physician NR	Univariate analysis: -Number of daily taken immunosuppressive pills: OR 1.33 (95% CI 1.04–1.69, *p* = 0.022) -Calcineurin inhibitors (CsA or FK) only: OR 5.513 (95% CI 1.17–25.86, *p* = 0.030) -Combination of immunosuppressors and steroids: OR 8.56 (95% CI 1.62–45.16, *p* = 0.011) -Number of daily taken concomitant medications: OR 0.87 (95% CI 0.77–0.99, *p* = 0.035) Multivariate analysis: -Higher number of daily taken immunosuppressive pills: OR 1.42 (95% CI 1.08–1.87, *p* = 0.011) -Lower number of daily prescribed concomitant medications: OR 0.85 (95% CI 0.74–0.98, *p* = 0.024)	-	Higher grades of cGvHD correlate with MNA: OR 3.01 (95% CI 1.27–7.14, *p* = 0.012) -No/mild cGvHD = 62.2% among adherent patients -Moderate/severe cGvHD = 80.2% among non-adherent patients	This is the first study to describe an association between cGvHD and MNA. A high prevalence of MNA was shown, particularly regarding timing and taking Surprisingly, patients taking a higher number of immunosuppressive agents as well as patients taking a lower number of concomitant medications were more likely to be non-adherent
Ice et al., 2020 [6], USA	Non-immunosuppressors = 200 (100.0%) Oral immunosuppressors (CsA, FK, sirolimus, prednisone, mycophenolate) = 153 (76.5%) Mean number of medications = 17 (3–42) Mean number of scheduled medications = 12 (1–29) Mean number of as-needed medications = 5 (0–19)	MMAS, a validated eight-item self-report questionnaire for non-immunosuppressors ITAS, self-report nine-item questionnaire validated in the kidney transplantation context for immunosuppressors Immunosuppressor serum drug levels (*n* = 29, 14.5%) Prescription refill records for immunosuppressors (1 − ([days between refills − total supply in days]/days between refills) × 100% (*n* = 15, 75.0%) High MA = 98 (48.0%) MNA = 102 (51.0%, 95% CI 44.1–57.9%) (MMAS) High MA = 95 (62.1%) MNA = 58 (37.9%, 95% CI 30.2–45.6%) (ITAS) No differences in MNA regarding drug monitoring (OR 0.98, 95% CI 0.94–1.02, *p* = 0.25) No differences in NMA regarding prescription refill records (*r* = −0.3015, *p* = 0.27) Patients and collected by nurses during clinical appointments and medical records Laboratory NR for the questionnaires, serum drug levels and prescription refill records collected for the previous 3 months	Univariate analysis: (MMAS) -Younger age (55.5 vs 57 years, *p* = 0.009) Multivariate analysis: (MMAS) -Older age (OR 0.97, 95% CI 0.94–0.99, *p* = 0.016) -Severe distress (OR 1.15, 95% CI 1.01–1.31, *p* = 0.035) using the Distress Thermometer, a validated tool -Higher number of as-needed medications (OR 1.14, 95% CI 1.03–1.27, *p* = 0.013) -Increased number of scheduled medications (OR 0.92, 95% CI 0.86–0.998, *p* = 0.043) (ITAS) -Distress (OR 1.17, 95% CI 1.02–1.34, *p* = 0.026) -Non-Caucasian race (OR 8.86, 95% CI 0.94–83.5, *p* = 0.057)	-	cGvHD (based on the ITAS) = 161 (80.5%) -Mild = 49 (24.5%) (OR 2.63, 95% CI 1.04–6.66, *p* = 0.042) -Moderate = 74 (37.0%) -Severe = 38 (19.0%) Mortality based on the MMAS = HR 1.43, 95% CI 0.83–2.44, *p* = 0.19; based on the ITAS = HR 1.33, 95% CI 0.72–2.44, *p* = 0.36. Due to: -GvHD (*n* = 16) -Relapse (*n* = 12) -Infection (*n* = 9) -Multiorgan failure (*n* = 5) -Unknown (*n* = 5) -Secondary malignancy (*n* = 4) -Cardiovascular disease (*n* = 3)	MNA to immunosuppressors was associated with mild cGvHD Moreover, MNA was found to be highly prevalent for both immunosuppressors and non-immunosuppressors
Mishkin et al., 2018 [11], USA	NR	Criteria for life-threatening NA based on a consensus by an interdisciplinary team, based on evidence from solid organ transplantation NA = 18 (21.0%) * NA among allogeneic (*n* = 42) = 13 (30.9%) Interdisciplinary team (five HSCT clinicians, namely physician assistants, nurse managers, social workers and oncologists) Electronic medical records after discharge	Univariate analysis: -Higher SIPAT, a validated psychosocial assessment tool: OR 1.162 (*p* < 0.001) Multivariate analysis: -Higher SIPAT: RR 4.98 (*p* < 0.0001) -Allogeneic HSCT: OR 14.184 (*p* = 0.005)	-	No association between adherence and survival [11]	A cut-off score of 18 provided optimal specificity (89.6%) and sensitivity (55.6%) for NA with SIPAT NA rates were 58.8% and 11.8% for subjects with SIPAT ratings of ≥ 18 or < 17, respectively (RR 4.98, *p* < 0.0001) Psychosocial risk as quantified by the SIPAT correlated with adherence to the post-transplant regimen. Moreover, high-risk SIPAT patients (OR 11.679, *p* = 0.002) and allogeneic respect to autologous (OR 6.867, *p* = 0.034) had an increased risk of being admitted to the ICU However, no correlation between SIPAT score or NA and morbidity, readmissions or mortality
Posluszny et al., 2018 [12], USA	Immunosuppressors twice/daily (NR) Non-immunosuppressors daily (NR)	Adapted version of the 16-item HHA (two items related to oral MA), a validated self-report assessment tool for both patients and caregivers Modified version of the item ‘Who was mostly responsible for this task being accomplished?’ from the Family Responsibility Questionnaire MA to immunosuppressors = 15 (71.4%) (90% among patients mostly responsible vs 50.0% among caregivers mostly responsible or shared responsibility, *p* = 0.063) MA to non-immunosuppressors = 16 (71.4%) (100% among patients mostly responsible vs 45.0% among caregivers mostly responsible or shared responsibility, *p* = 0.012) Dyads of patients and caregivers for self-report During follow-up	Immunosuppressors: -Better relationship quality as reported by the patients on the six-item QMI (*r* = 0.52, *p* = 0.015) -Less perceived stressfulness of the medical regimen on a dichotomous item from Lee et al. [43], by both the patients (r = −0.45, *p* = 0.042) and the caregivers (r = −0.57, *p* = 0.011) Non-immunosuppressors: -Less perceived stressfulness of the medication regimen by the caregivers (r = −0.46, *p* = 0.048)	-	-	Although adherence to attending medical appointments was 100%, adherence to all other tasks was not optimal Immunosuppressor MA was 71%; perceived regimen stressfulness appeared to be a more important factor than distress (depression and anxiety) in relation to medication taking Adherence levels for some tasks were influenced by which member of the dyad took responsibility for its accomplishment. Thus, strategies to improve adherence should consider dyadic factors including division of task responsibility
**Quasi-experimental studies**						
Charra et al., 2021 [7], France	Immunosuppressives: CsA = 22 (84.6%) FK = 4 (15.4%) (intervention group) CsA = 32 (91.4%) FK = 3 (8.6%) (control group)	Immunosuppressor drug serum levels in the therapeutic target range Number of serum assays MA = 61.5% (intervention group) vs 53.0% (control group) (*p* = 0.07) Mean number of serum assays = 11.5 (intervention group) vs 10.9 (control group) (*p* = 0.46) until 100 days post-HSCT or until immunosuppressor tapering Immunology laboratory Day care follow-up	-	Prospective cohort (*n* = 26): 79 pharmaceutical consultations (median, 3 per patient) -The day before discharge consisting of proactive medication reconciliation, personalised medication intake schedule, patient education and contact with community pharmacy (mean duration of 25 min) -During weeks 2 and 4 after discharge from the HSCT unit and once a month until day 100 post-HSCT, consisting of pharmaco-therapeutic analysis of prescriptions, review of medication with patient, identification of drug related problems at home, patient education (mean duration of 16 min) Retrospective cohort (*n* = 39): no pharmaceutical consultations, standard patient follow-up (not better specified)	-	Implementation of a specialised clinical pharmacy programme for patients who have received allogeneic HSCT seems to be beneficial for immunosuppressor adherence However, there were no significant differences with respect to aGvHD, infections and hospital readmission rates
Chieng et al., 2013 [24], Australia	Antiemetics (NR) Azole antifungals (NR) Ganciclovir (NR) Immunosuppressors (NR) Prophylactic antibiotics: -Sulfamethoxazole/trimethoprim (NR) -Phenoxymethylpenicillin (NR) Ursodeoxycholic acid (NR)	Adapted MMAS score, four-item questionnaire Number of dose administration aids Number of dispensation records Mean decrease of 1.53 points on the MMAS (95% CI 1.12–1.94, *p* < 0.0001) with score 0 (adherence = 100.0%) at the sixth visit (*n* = 17) Accurate use of aids and dispensation records Patients for self-report Clinical pharmacist Pharmacy dispensing system Weekly during each ambulatory visit from the second week post-discharge	-	Six weekly consultations (mean duration of 20 min, the first at week 2 post-discharge and the other within 7–10 days) by a clinical pharmacist with postgraduate qualification in clinical pharmacy and extensive experience in cancer care At every visit (*n* = 109), medication reconciliation interviews (*n* = 161, 1.4 per patient visit) were recorded and blindly assigned a risk rating by a multidisciplinary panel, considering the potential impact if the intervention had not occurred in combination with the likelihood of having to intervene in the same drug-problem in the future	-	40% of interventions were rated as high risk, 46% as medium risk and 14% as low risk; high- and medium-risk interventions constituted > 80% of the total MMAS scores improved by an average of 1.53 points (*p* < 0.0001) and all the patients were scored as highly adherent by the last visit
Polito et al., 2021 [25], Canada	NR	VAS MA = median 100% in both groups, interquartile range = 0 (intervention group) vs interquartile range = 5 (comparison group) (*p* = 0.12) Patients for self-report At a follow-up appointment 3–5 weeks post-discharge	-	Intervention group (SMP, *n* = 25): provision of medication counselling and medication charts by a pharmacist during the hospital stay at a time determined by the interprofessional health care team and supervision of patients’ self-administration of medications in a way similar to how it would be dispensed from a retail pharmacy by a nurse, who verifies the dose and documents that the dose has been taken by signing off on patient’s eMAR until discharge from HSCT unit Comparison group (*n* = 26): provision of a detailed one-on-one medication counselling session performed by a pharmacist within 24 h before discharge from the HSCT unit and of a personalised medication chart detailing the discharge medication schedule	-	Median knowledge scores in the comparison group vs the SMP group were 8.5/10 vs 10/10 at discharge (*p* = 0.0023) and 9/10 vs 10/10 at follow-up (*p* = 0.047). Differently, median self-efficacy scores at both discharge and follow-up were no significantly different (*p* discharge = 0.11, *p* follow-up = 0.10) The SMP did not result in a significant difference even in self-reported MA. However, the SMP was associated with at least 1 medication event in 7 participants, but no medication incidents occurred Patient and staff surveys showed a positive perceived value of the SMP
Zanetti et al., 2022 [26], Brazil	NR	BMQ, a validated 11-item self-report questionnaire for antihypertensive drugs Compliance (*n* = 27): 11 (40.74%) before medication advice and educational activities vs 19 (70.37%) until the end of the follow-up period (*p* = 0.0115) Compliance (*n* = 18 adults): 7 (38.9%) before medication advice and educational activities vs NR until the end of the follow-up period Patients self-report collected by pharmacist Before medication advice and educational activities and until the end of the follow-up period	No factors (age, gender, schooling, occupation, marital status, income, type of HSCT, source of stem cells, health problems or comorbidities) among the investigated before the first consultation *(p* > 0.05)	Pharmaceuticals consultations by a clinical pharmacist (*n* = 390) in the -Inpatient setting (from admission to discharge): daily pharmacotherapy review (analysis of the prescriptions, laboratory tests and patient’s clinical evolution, aiming at the prevention, monitoring, detection and resolution of DRPs (*n* = 278, 89.97%), production of educational materials on the use of medications for the health team, medication reconciliation at admission and discharge, discussion of discharge prescription and participation in the daily clinical team meetings -Outpatient setting (*n* = 130, 4.81 ± 1.80 per patient, lasting nearly 20 min): weekly pharmaceutical consultations consisting of pharmacotherapy review (*n* = 31, 10.30%), production of educational materials for the patient (using pictograms, medication organisers and booklets, guidelines on the use and storage of the medications) and other educational activities and participation in the clinical team meetings	Number of hospital readmissions at 100 days post-HSCT: χ^2^ = 7.816 (*p* = 0.021) GvHD: χ^2^ = 0.622 (*p* = 0.633) Transplant-related mortality: 0 (-)	Pharmacotherapeutic follow-up contributed to improve medication compliance (*p* = 0.0115) No relation between knowledge and compliance scores (*p* = 0.438) at the last consultation; thus, knowledge about pharmacotherapy alone does not translate into behaviours, which corroborates the complexity of the biopsychosocial factors associated with medication compliance
**Correlational study**						
Hoodin, 1993 [10], USA	NR	Number of medication infractions using the 24-hour recall method, a validated self-report method Pill counting Mean MA is the same among the three repeated measures (94.7% ± 7.8): *F* (2, 45) = 4.480, *p* = 0.017 -3.8–6.5% taking over-dose -8.3–11.1% taking under-dose -14.3–18.0% taking more of at least one dose -30.0–35.7% taking less of at least one dose MNA among allogeneic (*n* = 41) ≈ 7% Pill counting corroborated a mean of 67% (at second occasion) and 73% (at third occasion) Patients collected from staff during ambulatory visit and by telephone On three occasions between 84 and 100 days post-HSCT	None of the pre-HSCT predictors of affective distress were significant. Multivariate analysis: -Allogeneic (R^2^ = 0.145, *F* (5, 48) = 1.627, *p* = 0.171): *ß* = −0.303, t = −2.257, *p* = 0.029 -Self-reported pre-HSCT adherence based on SRPA, a validated nine-item self-report questionnaire (R^2^ = 0.145, *F* (5, 48) = 1.627, *p* = 0.171): *ß* = −0.083, t = 0.586, *p* = 0.561	-	-	Rate of mean hygiene adherence (65.2% ± 16.6%) was significantly worse than MA (94.7% ± 7.8%) or environmental restrictions (93.7% ± 6.9%) (*p* < 0.001), with autologous transplant patients committing more infractions than allogeneic recipients, despite their simpler medication regimen (*p* = 0.02) None of the demographic or psychological variables predicted medication or hygiene adherence. However, total inpatient days predicted hygiene adherence (*p* < 0.01) These findings indicate that although adherence is quite high to complex medication regimens and environmental restrictions, hygiene adherence is a significant problem, particularly for autologous patients
**Cohort studies**						
Scherer et al., 2021 [15], Germany	Immunosuppressors (NR)	MESI, a validated seven-item self-report questionnaire at 3, 12 and 36 months after HSCT A 5-point Likert scale, not validated before HSCT (*n* = 61) MA = 17 (40.5%), 3 months after HSCT (*n* = 42) MA = 14 (53.9%) 12 months after HSCT (*n* = 26) MA = 7 (50.0%) 36 months after HSCT (*n* = 14) Patients for self-report Before transplantation, at 3, 12 and 36 months after HSCT	TERS, a validated 10-item questionnaire: -Correlates with pre-HSCT MA assessed by clinicians on a 5-item Likert scale (mean score = 2, *r* = 0.36, *p* = 0.011) (*n* = 61) -Does not correlate with MESI, in the inpatient (*p* = 0.85, *r* = −0.026, 3 months after HSCT, *p* = 0.12, *r* = −0.206, 12 months after HSCT) and outpatient (*p* = 0.96, *r* = 0.009, 3 months after HSCT, *p* = 0.9, *r* = 0.03 12 months after HSCT) settings	-	-	Two groups emerged from the TERS score: the low-risk (26.5–29.0) and the increased-risk (29.5–79.5) groups. The increased-risk group showed significantly worse cumulative survival in the outpatient setting (log rank [Mantel Cox] *p* = 0.029] compared with the low-risk group, but there was no significant result for the interval immediately until 3 years after HSCT Pre-transplant screening with TERS contributes to predict survival after HSCT. The reason remains unclear because TERS scores did not correlate with GvHD or MESI
Posluszny et al., 2022 [9], USA	Immunosuppressors twice/daily (NR) Non-immunosuppressors daily (NR)	A modified version of the 17-item HHA (two items related to oral MA), a validated self-report assessment tool for both patients and caregivers (a single measure for each item was determined by comparing the responses of each member’s dyad) MNA to immunosuppressors = 10 (11.2%) at week 4 after discharge (*n* = 89) and 14 (15.7%) at week 8 after discharge (*n* = 84) MNA to non-immunosuppressors = 31 (34.8%) at week 4 after discharge (*n* = 89) and 33 (38.6%) at week 8 after discharge (*n* = 84) Patients and caregivers interviewed by a trained interviewer At week 4 and week 8 after discharge	Multivariate analysis: At 4 weeks post-discharge: (based on an adaptation of a tool validated in the heart disease population) -Lower caregiver task efficacy (*F* (8, 527), *p* = 0.004): *b* = −0.30, *p* < 0.01 At 8 weeks post-discharge: -Higher patients’ education level (*F* (5, 440), *p* = 0.002): *b* = 0.32, *p* < 0.01	-	-	NA rates varied among tasks, with 11.2–15.7% of the sample reporting MNA to immunosuppressors, 34.8–38.6% to other types of medications, 14.6–67.4% to required infection precautions, and 27.0–68.5% to lifestyle-related behaviours (as diet/exercise) NA rates were generally stable but worsened over time for lifestyle-related behaviours; the most consistent predictors were patient and caregiver pre-HSCT perceptions of lower HSCT task efficacy Higher caregiver depression, caregiver perceptions of poorer relationship with the patient, having a non-spousal caregiver, and diseases other than AML also predicted greater NA in one or more areas
**Studies reporting prevalence data**						
García-Basas et al., 2020 [8], Spain	Immunosuppressors (*n* = 46): -CsA = 46 (100.0%) -FK = 1 (2.2%) -Sirolimus = 3 (6.5%) -Mycophenolate = 34 (73.9%) Anti-infection prophylaxis (*n* = 19): -Antifungal (*n* = 13, 28.3%): -Posaconazole = 8 (61.5%) -Voriconazole = 4 (30.8%) -Posaconazole + voriconazole = 1 (7.7%) Antiviral with valganciclovir = 10 (21.7%)	Mean of dispensation records Reports on amounts of medication dispensed at different dates for mycophenolate, FK, sirolimus, posaconazole, voriconazole and valganciclovir CsA, FK and sirolimus serum level in the therapeutic range MA to immunosuppressors against secondary graft failure = 38 (82.6%) CsA = 39 (84.8%) Sirolimus = 3 (100.0%) MA to immunosuppressors against GvHD = 37 (80.4%) CsA = 39 (84.8%) Sirolimus = 3 (100.0%) Mycophenolate = 29 (84.2%) MA to anti-infection prophylaxis = 46 (100.0%) Pharmacy department Community pharmacy NR	-	-	Secondary graft failure: 0 (-) aGvHD = 17 (45.9%) in adherent patients vs 5 (55.6%) in non-adherence patients: OR 0.68 (95% CI 0.157–2.943, *p* = 0.718) Infections = 31 (67.4%) Readmission rates among adherent patients due to: -aGvHD (18.0%) -Infections (75.0%)	Acceptable adherence to prophylaxis has been seen against acute complications, although a considerable percentage of patients did not to take their medication as prescribed Correct adherence to immunosuppressors seems to reduce the risk of developing aGvHD
Lehrer et al., 2018 [32], France	CsA = 29 (87.9%) NR Mean number of prescribed drugs = 12	MMAS, a validated eight-item self-report questionnaire Poorly adherent = 18 (54.6%) Low MA = 2 (6.1%) Medium MA = 16 (48.5%) Highly adherent = 15 (45.4%) Patients NR	MA increased as age increased (*ρ* = 0.47, *p* = 0.03)	-	No association between MA and GvHD (26.7% vs 38.9%, *p* = 0.71)	More than half (54.6%) of patients were poorly adherent to medication The authors suggest that younger patients have poorer adherence than older patients

*b*: Non-standardised beta regression coefficient; *ß*: standardised beta regression coefficient; aGvHD: acute graft-versus-host disease; AML: acute myeloid leukaemia; BAASIS: Basel Assessment of Adherence to Immunosuppressive Medication Scale; BMQ: Brief Medication Questionnaire; CET: Compliance Evaluation Test; cGvHD: chronic graft-versus-host disease; CI: confidence interval; CsA: cyclosporine A; DRP: drug-related problem; eMAR: electronic medication administration records; *F*: Fisher’s exact test; FK: tacrolimus; GvHD: graft-versus-host disease; HHA: Health Habits Assessment; HR: hazard ratio; HSCT: haematopoietic stem cell transplantation; ICU: intensive care unit; ITAS: Immunosuppressant Therapy Adherence Scale; MA: medication adherence; MESI: Medication Experience Scale for Immunosuppressants; MMAS: Morisky Medication Adherence Scale; MNA: medication non-adherence; *n*: number; NA: non-adherence; OR: odds ratio; *p*: *p*-value; QMI: Quality of Marriage Index; *r*: Pearson’s correlation coefficient; RR: relative risk; *ρ*: Spearman’s correlation coefficient; SIPAT: Stanford Integrated Psychosocial Assessment for Transplantation; SMP: self-medication programme; SRPA: Self-Rating of Pre-BMT Adherence; TERS: Transplant Evaluation Rating Scale; VAS: visual analogue scale; vs = versus. * Including both autologous and allogeneic HSCT.

**Table 3 cancers-15-02452-t003:** Interventions delivered by clinical pharmacists and the related outcomes among the identified studies (*n* = 4).

Author, Year	Aims Related to MA	Population	Intervention(s)	Rates and Measures of MA	Effectiveness
Charra et al., 2021 [7]	To evaluate the impact of a specialised clinical pharmacy programme on adherence to oral immunosuppression treatment after discharge from the HSCT unit	*n* = 61 total *n* = 26 (42.6%) intervention group *n* = 35 (57.4%) control group	Prospective cohort (*n* = 26): 79 pharmaceutical consultations (median of 3 per patient) - The day before discharge (proactive medication reconciliation, personalised medication intakes schedule, patient education, contact with community pharmacy; mean duration of 25 min) - During weeks 2 and 4 post-discharge and once a month until day 100 post-HSCT (pharmacotherapeutic analysis of prescriptions, review of medication with the patient, identification of drug-related problems at home, patient education; mean duration of 16 min) Retrospective cohort (*n* = 39): no pharmaceutical consultations, standard patient follow-up (not specified further)	Drug serum levels in the therapeutic target range: 61.5% (intervention group) vs 53.0% (control group) (*p* = 0.07) Mean number of serum assays: 11.5 (intervention group) vs 10.9 (control group) (*p* = 0.46)	↕
Chieng et al., 2013 [24]	To evaluate the effectiveness of a specialty clinical pharmacist working in an ambulatory stem cell transplant clinic	*n* = 23	Six weekly consultations (the first at week 2 post-discharge and the other within 7–10 days; mean duration of 20 min) by a clinical pharmacist with a postgraduate qualification in clinical pharmacy and extensive experience in cancer care	Mean decrease of 1.53 points on MMAS (95% CI 1.12–1.94, *p* < 0.0001) with score 0 at the sixth visit (*n* = 17) Accurate use of dose administration aids and dispensation records	↑
Polito et al., 2021 [25]	To assess the impact of a self-medication programme on adherence	*n* = 51 total *n* = 25 (49.0%) (intervention group) *n* = 26 (51.0%) (comparison group)	Intervention group (*n* = 25): provision of medication counselling and medication charts by a pharmacist during the hospital stay and supervision of patients’ self-administration of medications by a nurse, who documents the taken dose by signing off on the patient’s eMAR until discharge Comparison group (*n* = 26): provision of a detailed one-on-one medication counselling session performed by a pharmacist within 24 h before discharge and of a personalised medication chart detailing the discharge medication schedule	Median 100% based on the VAS, interquartile range = 0 (intervention group) vs 5 (comparison group) (*p* = 0.12)	↕
Zanetti et al., 2022 [26]	To evaluate the results of pharmacotherapeutic follow-up on medication compliance and on the patients’ knowledge about pharmacotherapy	*n* = 27 total *n* = 18 (66.7%) adults *n* = 9 (33.3%) children	Pharmaceuticals consultations conducted by a clinical pharmacist (*n* = 390) in: - Inpatient setting (from admission to discharge): daily pharmacotherapy review (analysis of the prescriptions, laboratory tests and patient’s clinical evolution, aiming at the prevention, monitoring, detection and resolution of DRPs), production of educational materials on the use of medications for the health team, medication reconciliation at admission and discharge, discussion of discharge prescription and participation in the daily clinical team meetings - Outpatient setting (*n* = 130, 4.81 ± 1.80 per patient, lasting nearly 20 min): weekly pharmaceutical consultations consisting of pharmacotherapy review (prevention, monitoring, detection and resolution of DRPs), production of educational materials for the patient (using pictograms, medication organisers and booklets and guidelines on the use and storage of the medications) and other educational activities, participation in the clinical team’s meetings	11 (40.74%) based on the BMQ (before medication advice and educational activities) vs 19 (70.37%) (at the last consultation) (*p* = 0.0115) (*n* = 27) 7 (38.9%) based on the BMQ (before the first consultation) vs NR (for the last consultation) (*n* = 18 adults)	↑ but NR for the adult population

BMQ: Brief Medication Questionnaire; CI: confidence interval; DRP: drug-related problem; eMAR: electronic medication administration records; HSCT: haematopoietic stem cell transplantation; MA: medication adherence; MMAS: Morisky Medication Adherence Scale; n: number; NR: not reported; *p*: *p*-value; VAS: visual analogue scale; vs: versus. ↑: outcome improved; ↕: outcome not significantly different. At the overall level, only one study [24] yielded positive significant results for adults, even if the delivery of pharmacist-led interventions showed positive effects on MA [7,25,26].

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
