# Peer review of "Medication Adherence among Allogeneic Haematopoietic Stem Cell Transplant Recipients: A Systematic Review"

_cancers, 2023, doi:10.3390/cancers15092452_

Round 1

Reviewer 1 Report

Visintini et al. summarized the available oral MA prevalence data among adults allo-HSCT recipients and the tools used to measure it as well as predictors and risk factors of MNA, the effectiveness of interventions and the clinical outcomes of MNA. The manuscript is well-written.

 Comments

1. The manuscript should be more concise especially Discussion section.

2. Page 7 Line 308: The abbreviation of “cyclosporine A” should be “CsA”.

Author Response

Dear reviewer,

thank you very much for your interest in our review and your comments.

We considered your proposal to reduce and be more concise in the Discussion that has been shortened. In addition, we changed the abbreviation of “cyclosporine A” with CsA along the text.

Thank you

Reviewer 2 Report

The article is important and interesting for its comprehensive review of medication adherence (MA) in allogeneic hematopoietic stem cell transplant patients. It is also valuable in that it attempts to identify risk factors that may reduce MA.

1. Can the authors provide a mechanism by which higher education is associated with MNA? Is higher education associated with higher distress and higher psycochocial risk?

2. The fact that the analysis is of studies that are heterogeneous throughout, including the background factors of MA and its measurement, may make the interpretation of the systematic review difficult. Therefore, it is understandable that the authors gave up on the meta-analysis, but for this reason, the section at the beginning of page 26, Interventions to Improve MA, seems important. I feel that this section does not provide a clear conclusion.

3. I would like to suggest the authors to mention or discuss on the association between comorbidities and NMA.

Author Response

Reviewer,

Thank you very much for your consideration, feedback, and for having given us the possibility to revise the manuscript answering to the issues requested. We really appreciated it.

  1. We could not provide a possible mechanism regarding higher education and MNA, as it did not emerge from the study reporting this association (Posluszny et al., 2022). However, we considered your suggestion and added in the Discussion that literature is mixed on the role of education level in adherence in other diseases.

  1. Thank you, we consider your suggestion and added a phrase in the Result section to give more clarity. Moreover, we emphasized this concept in the Discussion section, adding a final sentence.

3. Thank you for this suggestion. Following the study protocol, we did not research for an association between comorbidities and MNA. However, we added it in the Limitations, considering that comorbidities could influence MNA.
